# Position: Topological Machine Learning Cannot Progress without Experimental Standards

Inés Castilla Rieso [1 2]    Samuel Peltier [1]    Philippe Carre [1]

## Abstract

Topological Machine Learning provides strong discriminative power for classification tasks through the use of Topological Data Analysis, and more particularly, Persistent Homology. Although it has strong theoretical appeal, it remains underused by the broader Machine Learning community; criticism often targets the reliance on synthetic data and the absence of shared experimental standards, which makes reported results difficult to compare. Indeed, current empirical evaluations lack a consistent framework for assessing methods: the construction of topological signatures is often opaque, statistical significance testing to validate reported gains, computing times and robustness to perturbations–such as missing data or noise–are often omitted. We assert that **progress in Topological Machine Learning depends on establishing clear and consolidated experimental standards that support meaningful comparison across methods**, articulated through a transparent and reproducible empirical framework including data processing and performance evaluation. We review current practices, highlight their limitations, and propose a set of principles for conducting rigorous and comparable empirical evaluations. Adopting these standards will enable trustworthy studies, clarify the gains of new methods, and ultimately support the broader adoption of Topological Machine Learning by the Machine Learning community.

## 1. Introduction

Topological Data Analysis (TDA) produces discriminative topological embeddings, which have proven particularly effective for classification tasks. This pipeline constitutes what we refer to as Topological Machine Learning (TML). TML's success, however, rests on a substantial amount of design choices and extensive processing. TDA also remains difficult to access for the broader Machine Learning (ML) community: the literature is often highly theoretical, and empirical evaluations lack transparency and methodological consistency, which we argue hinders future progress. Yet TML is set to remain a foundational paradigm of ML, therefore its empirical foundations require a more consolidated and standardized evaluation framework. Recent efforts have been made to unify the growing TML pipeline through comprehensive surveys and comparative studies (Barnes et al., 2021; Conti et al., 2022; Ali et al., 2023; Bandiziol & De Marchi, 2024), clarifying several components of the pipeline, and shedding light on recurring design choices. Yet, despite these advances, cross-study comparisons remain elusive, largely due to heterogeneous processing and evaluation practices. Indeed, choice of datasets matters, and from given dataset onward, successive processing steps may differ across experiments, making methodological alignment difficult. Moreover, computational costs are rarely reported in a step-wise manner, even though different design choices can dramatically affect runtime. While multiple runs are often performed to assess the robustness of the results, the number of runs varies widely across studies, and statistical tests assessing the significance of performance superiority are seldom reported.

> **Position:** the absence of a consolidated experimental framework undermines the interpretability of empirical results in TML.
>
> We need transparent standards for TML to progress within the broader ML community, this article aims to:
>
> **(i)** Identify the methodological blind spots of current TML practice.
>
> **(ii)** Examine how these choices influence outcomes and assess the validity of cross-study comparisons.
>
> **(iii)** Establish a standard experimental framework for TML.

[1]Université de Poitiers, Univ. Limoges, CNRS, XLIM, Poitiers, France [2]CHU Poitiers Clinical Investigation Center CIC 1402, Université de Poitiers, INSERM, Poitiers, France. Correspondence to: Inés Castilla Rieso <ines.castilla@univ-poitiers.fr>.

*Proceedings of the 43rd International Conference on Machine Learning*, Seoul, South Korea. PMLR 306, 2026. Copyright 2026 by the author(s).

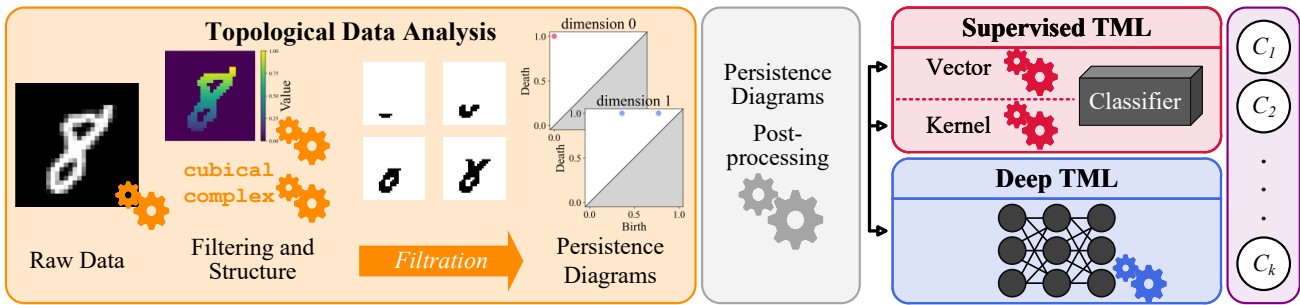

*Figure 1.* Outline of the Topological Machine Learning pipeline; the cogwheel icons highlight the steps where methodological choices may vary across studies. Topological Data Analysis enables the transformation of raw data into persistence diagrams: raw data first undergo up to three processing steps before a *filtration* is computed; additional post-processing can be applied to the diagram after its computation. Then, supervised TML relies on vector or kernel representations combined with classical classifiers, while deep TML uses neural architectures that learn directly from persistence diagrams.

**This position paper argues that progress in Topological Machine Learning depends on establishing clear and consolidated experimental standards that support meaningful comparison across methods.** The growing TML literature, and more recently the integration of deep architectures into topological pipelines (Carrière et al., 2020; Kim et al., 2020; Reinauer et al., 2021; Wang et al., 2024), makes this need more pressing. Neural networks require transparent and well-specified methodological protocols to ensure that reported gains in accuracy or runtime genuinely reflect algorithmic improvements rather than experimental variability. This concern has been recently recognized in the broader ML community: in 2019 the Neural Information Processing Systems (NeurIPS) conference introduced a reproducibility program aimed at improving how ML research is conducted, communicated, and evaluated (Pineau et al., 2021). Similar calls for methodological rigor have been articulated in meta-studies on Artificial Intelligence (AI) reproducibility (Gundersen & Kjensmo, 2018), underscoring the need for clearer standards and reporting practices. In this work, we use the term of ***experimental standards*** to refer to a minimal set of methodological requirements that would make TML studies comparable and reproducible. These include (i) clear pipeline reporting practices, (ii) end-to-end timing reporting, (iii) basic statistical testing to assess the significance of performance protocols. In the absence of such standards, empirical claims remain difficult to interpret, or compare. This paper is intended as a dialogue between the TDA and ML communities, with the goal of converging toward a standard experimental framework. We believe that such framework will support the integration of TML into the broader ML community.

To support our position, our contributions are threefold. First, in Section 2, we highlight key limitations of current empirical practices in TML, by identifying the steps that rely on engineering choices and vary across studies. Second, in Section 3, we examine how these choices shape empirical outcomes (performance, stability, and computational cost) and evaluate under what conditions results obtained in different studies can be meaningfully compared. Third, in Section 4, we outline a standard experimental framework for TML, articulating principles for transparent reporting, reproducible evaluation, and statistically grounded performance assessment.

## 2. Background

Topological Machine Learning combines Persistent Homology (PH) (Edelsbrunner et al., 2002; Zomorodian & Carlsson, 2005), a key tool of Topological Data Analysis (Carlsson, 2009), and Machine Learning algorithms for classification tasks. The TML pipeline consists of two successive steps: the construction of *persistence diagrams* from raw data through PH, followed by the embedding of these diagrams into *topological signatures*. The TML pipeline is described in Figure 1.

PH summarizes the evolution of connected components (dimension 0), loops (dimension 1), and higher-dimensional cavities (informally known as "holes") in data. A nested sequence of objects built by gradually increasing a scale parameter and tracking the appearance and disappearance of topological features, is a process called *filtration*. These appearance and disappearance of topological features across scales can be represented into a Persistence Diagram (PD): a multiset of points in $\mathbb{R}^2$, where each point $(b, d)$ records the scales at which a topological feature appears (*birth*) and disappears (*death*). The position of these $(b, d)$ points reflects the lifetime of the corresponding topological features.

Due to their multi-scale and multi-dimensional construction, the metric space of persistence diagrams is none-Hilbert,

which made their use in traditional ML algorithms challenging in the early days of TDA. This difficulty has motivated a decade of work on embedding persistence diagrams into Hilbert spaces (Euclidean or Reproducing-kernel) to make them compatible with standard ML classifiers. TML methods divide into two paradigms. **Supervised TML** builds explicit vector (Conti et al., 2022; Ali et al., 2023) or kernel (Bandiziol & De Marchi, 2024) representations of persistence diagrams and couples them with standard ML classifiers, e.g., Support Vector Machines (SVM), Random Forest (RF), Ridge, or kernel SVMs. **Deep TML**, by contrast, leverages deep architectures, e.g., Deep Sets (Zaheer et al., 2017), Transformers (Vaswani et al., 2017), that learn directly from persistence diagrams, aiming to integrate topological information more seamlessly into end-to-end learning pipelines (Hofer et al., 2017; 2019; Carrière et al., 2020; Reinauer et al., 2021; Wang et al., 2024). A similar approach relies on building differentiable topological layers from vector representations of persistence diagrams, which are also integrated into deep architectures (Kim et al., 2020). This line of work occupies an intermediate position between supervised and deep TML by being integrable into deep architectures while still depending on explicit persistence diagram vectorizations.

A key property of TDA is that it is not restricted to a specific type of data: PH can be computed on any input data such as point sets, images, graphs, meshes, and time series, even when samples differ in size. Unlike standard ML classifiers such as SVM, RF, ..., and neural architectures such as Fully Connected Networks (FCNs), Convolutional Neural Networks (CNNs), or Graph Neural Networks (GNNs)–which typically impose structural constraints on their inputs and may require regularization, or resampling–TDA naturally processes irregular, noisy, or incomplete data. By extracting persistence diagrams that are invariant to sample size and robust to perturbations, TDA offers a flexible and unified way to represent heterogeneous data, which contributes to the appeal of TML.

## 3. Limitations of Current Empirical Practices in TML

Building on this background, a thorough examination of the main shortcomings of current empirical practices in TML will serve to assess how methodological inconsistencies limit the interpretability and comparability of empirical evaluations.

### 3.1. Variability in Design Choices

We note that, software ecosystems used for TDA vary across studies. Implementations rely on a wide range of publicly available TDA libraries–including `Scikit-TDA` (Saul & Tralie, 2019) (`ripser`), GUDHI (Project, 2025),

`Giotto-TDA` (Tauzin et al., 2021), `Teaspoon` (Khasawneh et al., 2025), `Dionysus` (Morozov, 2025), ... Despite the heterogeneity in experimental settings, the software choices are typically well documented in the studies.

#### 3.1.1. DATA AND TML

One of the limitations of current empirical practice in TML is the absence of consensus on which datasets should serve as benchmarks. Existing studies rely on a heterogeneous collection of data sources, ranging from different versions of the same synthetic dataset to domain-specific real-world applications and proprietary collections. Although TML has been applied in practical settings (Majumdar & Laha, 2020; Smith et al., 2021; Skaf & Laubenbacher, 2022), where topological signatures are known to be informative, most methodological contributions evaluate their performance on synthetic and real-world benchmarks. As summarized in our survey table (Appendix A.1), a small number of benchmarks recur across the literature, generally organized into point-clouds datasets (Hertzsch et al., 2007; Sonego et al., 2007), image benchmarks (LeCun et al., 1998; Xiao et al., 2017; Ojala et al., 2002), and graph- or mesh-based collections (Pickup et al., 2014). Parameters governing data generation vary from one study to another, and some works rely on sub-sampled or size-restricted variants of the same dataset. This variability undermines reproducibility and prevents meaningful comparison across methods. While the reliance on synthetic data and real-world benchmarks is well defined, the real challenge lies in the coexistence of different data types evaluated under varying processing pipelines.

#### 3.1.2. FILTERING AND DATA TYPES

The second source of methodological variability in TML arises from the choice of *filtering functions*. Filtering plays a crucial role in shaping the topological structures that will be captured by the filtration: it transforms heterogeneous raw inputs into scalar fields whose sublevel sets reveal specific properties. Different data types naturally call for different filtering functions, each emphasizing particular aspects of the underlying structure (e.g., intensity patterns in images (Guo et al., 2010; Garin & Tauzin, 2019; Conti et al., 2022), spatial organization in point clouds (Chazal et al., 2011), or heat-kernel-based scalar fields in meshes and graphs (Sun et al., 2009), ...). As summarized in Appendix A.2, the literature exhibits a wide range of filtering choices, often tailored to the data type and task at hand. This diversity, while methodologically rich, also introduces substantial variability in empirical evaluations, since the selected filtering function directly influences the resulting persistence diagram and, ultimately, the performance of TML methods.

### 3.1.3. CHOICE OF STRUCTURE AND FILTRATION

The third source of methodological variability in TML is linked to the interplay between the *structure* imposed on the filtered data and the *filtration* used to extract topological information. Before computing PH, filtered data must be endowed with an appropriate combinatorial structure, such as geometric simplicial complexes for point clouds, cubical complexes for images, or intrinsic graph- or mesh-based structures for geometric objects. This structural choice determines the space on which the filtration operates and directly influences the topological features that can be detected. Filtrations themselves may be geometric (Chazal et al., 2014a), when driven by a scale parameter on a simplicial complex, or functional, when induced by a scalar function. In practice, different data modalities call for different structural assumptions and filtration strategies, and the literature is met with heterogeneous choices. Studies differ in the type of complex used (e.g., Vietoris-Rips, Čech, or Alpha (Edelsbrunner & Harer, 2010), cubical (Kaczynski et al., 2004), or mesh-based). This diversity, while methodologically rich, introduces variability in persistence diagrams, since both the structure and the filtration jointly determine the topological features extracted from the data.

### 3.1.4. PERSISTENCE DIAGRAM POST-PROCESSING

A fourth, optional, step appears in the TML pipelines that typically serves two distinct purposes. In kernel-based supervised or deep TML settings, where models can be computationally heavy, it often involves **reducing** the persistence diagram by removing points or dimensions, with the aim of lowering computational cost. In contrast, other works use this step to **aggregate** information by constructing *multi-vectors*, that is, feature representations obtained by combining several persistence diagrams derived from different filtering functions (Adcock et al., 2016; Reininghaus et al., 2015; Chung & Lawson, 2022; Conti et al., 2022) or from globally different processing choices (Garin & Tauzin, 2019). While these two practices address different objectives–computational efficiency on the one hand, and higher performance on the other–they both introduce additional variability in empirical pipelines, as these strategies vary widely across studies. Appendix A.3 synthesizes some end-to-end TML pipelines found in the literature: from raw data to post-processing; providing a unified overview of the methodological differences across studies.

### 3.1.5. SUPERVISED AND DEEP TML

Finally, supervised TML, vector- and kernel-based approaches, and deep architectures rely on distinct mechanisms for transforming persistence diagrams into representations suitable for learning. Vectorization methods produce fixed-length vectors (Chazal et al., 2014b; Adcock et al.,

2016; Bubenik, 2015; Di Fabio & Ferri, 2015; Adams et al., 2017; Kališnik Verovsek, 2019; Polanco & Perea, 2019; Royer et al., 2021; Chung & Lawson, 2022; Perea et al., 2023), kernel approaches yield similarity measures (Reininghaus et al., 2015; Kusano et al., 2016; Carrière et al., 2017; Le & Yamada, 2018), both can be combined with classical ML models. Deep architectures learn task-specific representations directly from the persistence diagrams (Hofer et al., 2019; Carrière et al., 2020; Reinauer et al., 2021). These families of methods therefore generate distinct outputs in classification tasks, with varying levels of performance and computational cost. A detailed comparison of these approaches lies beyond the scope of this position paper; our focus is on the empirical practices that govern how such methods are evaluated, rather than on the methods themselves.

## 3.2. Variability in Performance Evaluation

Beyond the diversity of pipeline configurations, empirical studies in TML also differ substantially in how performance is evaluated. Even when identical datasets and methods are used, the choice of evaluation metrics, the number of independent runs, the reporting of computational costs, and the treatment of perturbation or robustness tests vary widely across the literature. This lack of uniformity in evaluation protocols complicates cross-study comparison and obscures the practical reliability of TML methods. In this section, we review these sources of variability and highlight their impact on the interpretability of empirical findings.

Performance evaluation practices differ greatly from one study to another. Studies differ in the number of runs, ranging from single evaluations to 100 repeated trials, and the train/test partitions, as shown in Appendix A.4.1. Evaluation metrics are equally heterogeneous, with some works reporting accuracy alone, or with standard deviations, others providing balanced accuracy, precision/recall curves. Moreover, a systematic comparison with baseline or reference models is rarely included, making it difficult to assess whether observed performance differences come from genuine methodological improvements or from uncontrolled variability in the evaluation protocol.

Reporting of computational costs in TML studies is largely restricted to the vectorization or kernel-based stages, while the earlier components of the pipeline (filtering, structural construction, post-processing) are rarely accounted for. As we will examine in the next section, these choices can substantially alter the computational cost of the entire pipeline, yet they remain mostly unreported. Robustness analyses are similarly scarce. Appendix A.4.2 shows a selection of well-known TML studies that report computational costs or robustness evaluations. This overview is not intended to be exhaustive, but rather illustrate representative examples

of current practices. This highlights the fragmented and inconsistent nature of current practices.

# 4. Evaluating the Impact of Pipeline Choices on TML Performance

This section provides an empirical assessment of how the different steps of the TML pipeline influence both the computational cost of PH and the behavior of learning methods. All experiments were performed on a Windows 10 (64-bit) machine with an Intel® CPU (8 cores, 16 threads) and 31.2 GB of RAM, using Python 3.10.16 (CPython). `ripser` was used for Vietoris-Rips complexes, and `gudhi` was used for other complexes. We examine how design choices affect the size and structure of the resulting persistence diagrams (§4.1), and the downstream accuracy (§4.2), then, how PH computation scales across data modalities (§4.3), before concluding with a discussion on how these observations motivate the need for experimental standards (§4.4).

## 4.1. Design Choices and Diagram Complexity

To examine how design choices influence the size and structure of persistence diagrams across different types of data, we consider each modality separately. For this purpose we rely on one benchmark dataset per modality: **Orbits** for point clouds, **MNIST** for images, and **SHREC14** for graph and meshes. Each of which appears recurrently in the TML literature; as shown in Appendix A.1, they are the most frequently used benchmarks for their respective data modalities.

In this analysis, diagram size is quantified by the number of points (topological features) it contains, while structural complexity is assessed through Atienza's Entropy measure (2018).

### 4.1.1. POINT-CLOUDS: THE EXEMPLE OF ORBITS

To illustrate the impact of design choices on persistence diagram for point-cloud data, we rely on the Dynamical System Dataset, or Orbits dataset, which consists of point-clouds of fixed size. Each sample contains 1000 points in

$\mathbb{R}^2$ and detailed description of its construction is provided in Appendix B.1.

For point-cloud inputs, several geometric complexes tend to differ depending on the TML paradigm. In Deep-TML architectures, Alpha complexes are commonly used, as in the works of Carrière et al. (2020) and Reinauer et al. (2021). In contrast, supervised TML approaches typically rely on Vietoris-Rips complexes. Kim et al. (2020) used DTM-Rips complexes, the latter relying on measure-based filtering functions such as the Distance-to-Measure (Anai et al., 2020). Although these constructions are theoretically related, they differ substantially in the size and structure of the resulting persistence diagrams, as reported in Table 1. Alpha complexes yield the fastest computations and the largest number of $H_1$ points, while Rips and DTM-Rips progressively increase computational cost and produce increasingly sparse diagrams. Entropy values also vary, with Alpha and Rips showing lowest Entropy in dimension 0 and dimension 1, respectively. Post-hoc Dunn tests confirm that these differences are statistically significant for every pairwise comparison, across the number of $H_1$ points and entropy in both dimensions.

These results highlight that choice of configuration for point clouds has a pronounced and quantifiable impact on both the computational cost and the structural properties of the resulting persistence diagrams, in particular through the induced distribution of points across the diagram (Bobrowski & Skraba, 2023).

### 4.1.2. IMAGES: MNIST DATASET

To evaluate the TML pipeline on images, MNIST, a benchmark widely used both in TML and by the broader ML community, was used in a controlled setting. Each sample of the dataset consists of a $28 \times 28$ grayscale image representing a handwritten digit.

Most authors construct their topological representations by aggregating multiple persistence diagrams, as described in previous sections and Appendix A.3. The resulting representations differ in the number of persistence diagrams aggregated (combining dimensions 0 and 1) and in the computation time, as reported in Table 2.

*Table 1.* Average per-diagram computation time (in seconds), mean number of dimension 1 points (# $H_1$), and mean±std entropy in dimensions 0 and 1 on Orbit point clouds. Mean number of dimension 0 points is omitted because this number is stable (1000) across filtrations.

|  | TIME (S) | # $H_1$ | ENTROPY | |
|---|---|---|---|---|
|  |  |  | $H_0$ | $H_1$ |
| ALPHA | 0.01 | 898 | 5.1±0.8 | 5.2±0.8 |
| RIPS | 0.38 | 240 | 6.7±0.5 | 4.9±0.7 |
| DTM-RIPS | 173 | 182 | 6.6±0.5 | 4.5±0.7 |

*Table 2.* Number of aggregated persistence diagrams (# PD) and computation time (in millisecond) for different processing strategies on MNIST digit images.

| FILTERING | TIME (MS) | # PD |
|---|---|---|
| HEIGHT (H) | 6 | 8 |
| MEDIAN CANNY DISTANCE (MCD) | 3 | 2 |
| HEIGHT RADIAL DENSITY (HRD) | 25 | 36 |
| CLBP | 4 | 4 |
| INVERSED (I) | 3 | 4 |

Results highlight variability across methods: approaches based on multi-filterings (e.g., H or HRD) produce a large number of diagrams and consequently incur higher computation times. In contrast, methods relying on a single filtering (e.g. MCD) yield more compact representations and lower computational cost. Beyond the number of diagrams, the average number of points per diagram and the entropy of the resulting representations also differ significantly across methods. These additional statistics, reported in Appendix B.2, further illustrate how filtering choices shape the structural complexity of persistence diagrams, even when applied to images of identical resolution.

### 4.1.3. GRAPHS AND MESHES: SHREC14

Graphs and meshes are also often used in the TML literature, their underlying structure is no longer a regular grid, as for images, or an unstructured point set but a combinatorial or geometric network. In this graph-based setting, filtrations are typically built from scalar functions defined on nodes or edges. The choice of these scalar functions can alter the number of persistence points and the overall complexity of the resulting diagrams.

To evaluate the graph-based setting SHREC14, a widely used benchmark of 3D shapes, was used. Each mesh comes with a fixed vertex-face structure, and filtrations are commonly constructed from vertex-based scalar functions, among which the Heat Kernel Signature (HKS) (Sun et al., 2009) is a standard choice. HKS induces a filtration that reflects intrinsic geometric properties of the surface, and its behavior depends on both the mesh resolution and the parameterization of the HKS (e.g., number of time scales, or number of eigenvalues). As a result, the size and structural characteristics of persistence diagrams produced in dimensions 0 and 1 can vary across configurations.

*Table 3.* Average per-diagram computation times (in seconds), mean number of dimension 0 points (# $H_0$), and mean±std entropy in dimensions 0 and 1 on SHREC14 meshes, for different HKS parameters. Mean number of $H_1$ points is omitted because this number is stable (25) across HKS parameters. Entropy in $H_0$ is reported in units of $10^{-3}$.

| HKS | | TIME | # $H_0$ | ENTROPY | |
|---|---|---|---|---|---|
| T | EIG | (S) | | $H_0$ ($10^{-3}$) | $H_1$ |
| 1 | 50 | 4.71 | 6 | 5.3±3.1 | 1.6846±0.0905 |
| 1 | 100 | 8.44 | 6 | 5.3±3.1 | 1.6846±0.0905 |
| 1 | 200 | 20.42 | 6 | 5.3±3.1 | 1.6846±0.0905 |
| 5 | 50 | 4.83 | 29 | 4.7±1.5 | 1.6836±0.1330 |
| 5 | 100 | 8.27 | 29 | 4.7±1.5 | 1.6836±0.1330 |
| 5 | 200 | 20.04 | 29 | 4.7±1.5 | 1.6836±0.1330 |
| 10 | 50 | 4.73 | 34 | 4.9±1.4 | 1.6824±0.1377 |
| 10 | 100 | 8.10 | 34 | 4.9±1.4 | 1.6824±0.1377 |
| 10 | 200 | 20.13 | 34 | 4.9±1.4 | 1.6824±0.1377 |

HKS parameters have an influence on both the size and structure of the resulting persistence diagrams, as shown in Table 3. Across all configurations, the mean number of $H_1$ points remains stable. The mean number of $H_0$ points and entropy, both in $H_0$ and $H_1$, vary markedly with the time parameter $t$: larger values of $t$ produce more dense diagrams, and varying entropies. In contrast, the eigenvalue cutoff leads to longer runtimes. These results show that reducing the eigenvalue cutoff yields diagrams with the same size and comparable entropy, while significantly lowering computational cost. Statistical analyses (post-hoc Dunn tests) indicate that the time parameter $t$ has a statistically significant effect on both the size of the persistence diagrams (in $H_0$) and their entropy, confirming that variations in $t$ systematically alter the structure of the resulting persistence diagram.

### 4.2. Design Choices and Downstream Accuracy

Beyond their impact on computational cost and structural variability, design choices also strongly affect downstream accuracy. Table 4 reports downstream accuracy obtained under different methodological design choices across the three datasets (Orbits, MNIST and SHREC14). For each dataset, we consider two parameter-free vectorization methods–Persistence Statistics and Carlsson Coordinates–to avoid confounding effects due to hyperparameter tuning. For each configuration, we fix the classifier to a RF with 100 trees and evaluate accuracy over 100 runs on random 70/30 splits. This setup isolates the influence of design choices (filtration, structure, and post-processing) on performance. For clarity, we reported only a subset of design choices: for Orbits we excluded DTM-Rips due to its computation times, while for MNIST, we retained the configurations that yielded consistent accuracy values.

Even under controlled settings, both vectorization methods produced markedly different performance results, and best accuracy was not consistently associated with any specific design choice. We also performed statistical tests to identical design choices, resulting in 93.75% of p-values< 0.05. These tests confirm that, even when the processing steps are the same, accuracy differences between vectorization methods remain statistically significant in many cases. However, for the same dataset and fixed classifier/splits, best accuracy varies with the chosen pipeline, and no design choices emerge as universally optimal.

### 4.3. Scaling of Persistence Computation Across Data Modalities

Section 4.1 showed that computation times vary across data modalities and design choices. To understand the origin of this variability, we examined the different steps of the pipeline separately. In practice, a large fraction of the com-

*Table 4.* Mean accuracy (%) over 100 runs of a fixed Random Forest (100 trees) on 70/30 splits, for different design choices across datasets.

| | | PERSISTENCE STATISTICS | | | | CARLSSON COORDINATES | | | |
|---|---|---|---|---|---|---|---|---|---|
| DATASET | DESIGN CHOICES | $H_0$ | $H_1$ | $H_0+H_1$ | $H_0H_1$ | $H_0$ | $H_1$ | $H_0+H_1$ | $H_0H_1$ |
| ORBITS | ALPHA | 81.85 | 88.15 | **88.53** | 87.93 | 74.64 | 78.96 | 78.33 | 75.87 |
| | RIPS | 83.24 | 87.31 | 86.52 | 84.60 | 74.24 | **88.43** | 86.76 | 83.67 |
| MNIST | H | 61.17 | 76.21 | 82.95 | 83.91 | 42.79 | 79.17 | **85.99** | 82.83 |
| | HRD | 67.30 | 84.19 | **86.87** | 77.25 | 74.24 | 80.86 | 84.06 | 83.93 |
| SHREC14 | HKS ($t$=1) | 70.36 | 79.03 | **80.54** | 76.87 | 50.29 | 43.94 | 60.14 | 44.18 |
| | HKS ($t$=5) | 37.27 | 88.02 | 87.67 | 86.66 | 15.52 | **88.53** | 87.62 | 88.04 |
| | HKS ($t$=10) | 51.90 | **88.19** | 86.49 | 87.18 | 49.99 | 63.39 | 61.72 | 63.56 |

putational cost arises from the construction of the filtration, whose complexity depends heavily on the chosen structure.

To understand these effects, we first study how the cost of PH scales with the size of the input data in a controlled setting. This experiment isolates the intrinsic computational behavior of the most common filtrations used in TML pipelines. The computational cost of PH is fundamentally governed by the size and structure of the object on which the filtration is applied. Whether the input is a point cloud, an image, a graph, or a mesh, the number of elements involved in the filtration (points, pixels, or vertices) directly determines the size of the underlying complex and, consequently, the time required to compute persistence diagrams.

For point-cloud data, we generate synthetic Gaussian point sets of increasing resolution $n$, sampled as $x \sim \mathcal{N}(0, I_2)$. In practice, the points lie approximately within the interval

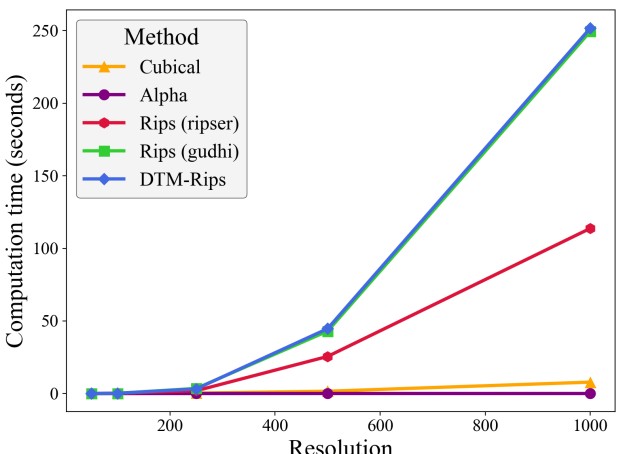

*Figure 2.* Scaling of persistence diagram computation for different complexes: cubical or geometric on Gaussian images or point clouds, respectively. The curves report the computation time $t(n)$ as a function of the resolution $n$ of the input image ($n \times n$) or point cloud ($n$ points).

$[-3, 3]^2$. We measure the computation time $t(x)$ required to compute the corresponding filtrations. For images inputs, we generate synthetic Gaussian-noise grids of increasing resolution $n \times n$ and compute persistence diagrams. Figure 2 shows how resolution influences the cost of PH, regardless of other processing steps.

The results show clear differences in computational cost across filtrations. Cubical and Alpha filtrations are the least expensive, whereas DTM-Rips is the slowest. However as noted in Section 4.1.1, Alpha filtrations produce more points, which can increase the cost of downstream embedding in supervised or deep TML. Indeed, prior studies, see Appendix A.4.2, indicate that the computational cost of topological signatures is strongly driven by the number of points in the persistence diagram.

### 4.4. Discussion and Implications

The analysis conducted across data modalities reveal a consistent pattern: the computational burden of TML pipelines is shaped by multiple interacting factors, and not solely by the PH computation itself. While the construction of persistence diagrams already exhibits substantial variability, driven by the choice of filtration, the structure of the underlying data, and the size of the resulting complexes, the post-processing stage introduces further constraints. Representation methods, whether based on vectorization, kernels, or deep architectures, scale directly with the size and number of persistence diagrams, often in ways that are computationally prohibitive. As a result many authors simplify or compress their diagrams, either by subsampling points, discarding homological dimensions, or applying heuristic reductions.

Although classical TDA commonly interprets long-persistence features as capturing meaningful topological structure (Cohen-Steiner et al., 2007), this interpretation does not directly extend to TML: the discriminative power of shorter persistence features remains unclear. Similarly, no principled hierarchy exists across homological dimen-

sions: an in-depth analysis is required (see Section 4.2) to determine which dimensions contribute most to downstream accuracy.

Taken together, these observations reinforce our central position: control over the experimental setup and design choices is essential to draw reliable conclusions about the performance of a given TML method. In particular, the empirical results presented in this section demonstrate that design choices do affect downstream accuracy, further motivating the need for standardized, transparent, and reproducible experimental practices in TML.

# 5. Call to Action: Establishment of Experimental Standards

The analyses presented in the previous sections reveal a central challenge for the TML community: the computational and methodological variability introduced at every step of the pipeline is substantial, and often under-reported. Together, all these observations show that the computational burden of TML pipelines is not limited to the construction of persistence diagrams. Understanding these costs is therefore essential for establishing realistic and reproducible experimental standards in TML.

These findings motivate a call for clear, actionable, and widely adoptable standards. The goal is not to restrict methodological creativity, but to ensure that comparisons across studies are meaningful, fair, and reproducible. Below, we outline a set of concrete principles that address the main sources of variability identified in our analysis.

## 5.1. Explicit Reporting of TML Pipeline and Systematic Reporting of Computation Times

A first requirement is the reporting of both hardware specifications and software information for all experiments. This would ensure that timing can be interpreted and compared meaningfully. A second requirement is the transparent documentation of all design choices involved in the TML pipeline. This includes, but is not restricted to:

- description of the dataset and parameters–for synthetic ones;

- definition of the filtering function and specification of its parameters;

- specification of the structure type;

- identification of any reduction, simplification, or aggregation applied to the persistence diagrams;

- characterization of the embedding method.

Because these choices directly affect the structure, size, and computational cost of the resulting topological signatures, **comparisons should only be made between identical pipelines**. For instance, methods built on different design choices operate on different underlying distributions, and comparing a method using a single filtration to one aggregating dozens of filtrations is misleading unless the computational costs are explicitly matched or reported. Clear reporting ensures that differences in performance can be attributed to methodological innovations rather than preprocessing decisions. We therefore recommend that authors systematically report computation times for full end-to-end pipeline and not only on the signature.

These measurements should be reported under a fixed hardware configuration and dataset size, or normalized appropriately. Such reporting is essential for assessing the practical feasibility of TML methods, especially in settings where diagram sizes vary. Moreover, without such information, it is impossible to evaluate whether a method is scalable, or whether its performance gains come at the cost of prohibitive computation.

## 5.2. Unified Evaluation Metrics and Statistical Comparisons

To ensure fair comparisons with non-topological baselines and across TML methods, we advocate for the use of unified evaluation metrics. Mean classification accuracy (balanced) or F1-score (precision or recall) should be reported consistently, depending on the task, on a sufficient number of runs. Beyond point estimates, performance should be assessed using statistical tests that quantify whether observed differences are meaningful and statistically significant.

For classification tasks, given two TML methods, classical tests include the McNemar test to test error patterns (Dietterich, 1998), and paired t-test or none-parametric Wilcoxon signed-rank test on performance distributions (Demšar, 2006). These statistical tools ensure that comparisons are not driven by random fluctuations or datasets splits, and they provide a principled basis for claiming improvements over existing methods.

## 5.3. Implementation Pathways

Several community initiatives have already demonstrated the value of coordinated efforts around TDA and its interaction with ML. Notable examples include the Topological Data Analysis and Beyond workshop at the 2020 NeurIPS conference[1], as well as the Special Session on Topological Data Analysis in Machine Learning held at the 18th IEEE International Conference on Machine Learning and Appli-

---

[1]https://tda-in-ml.github.io/

cations[2]. Similar Workshops or Special Sessions could be organized in the upcoming years to reflect on the evolution of TDA within modern ML pipelines (TML): integrating (i) experimental standards, (ii) insights from recent surveys, (iii) dialogue around deep-TML approaches. Such recurring community-driven efforts would naturally support the development and adoption of shared experimental standards, and further facilitate the integration of TML in the broader ML community.

## 6. Alternative Views

While this paper argues for the establishment of experimental standards in Topological Machine Learning, it is important to acknowledge that alternative perspectives exist regarding both the necessity and the form of such standards. Researchers may reasonably contend that the diversity of TML pipelines is a strength rather than a limitation, enabling methodological creativity and adaptation to domain-specific constraints. From this viewpoint, enforcing common reporting practices or unified evaluation protocols could be seen as restricting innovation or imposing unnecessary overhead on exploratory work.

Another perspective emphasizes that the computational variability we highlight may not always be problematic in practice. For instance, TML is still a young field, and heterogeneity in experimental setups is natural at this stage of development. According to this view, premature standardization might risk locking the community into conventions that later prove suboptimal. In certain applications, the choice of filtration or representation is guided primarily by domain knowledge rather than computational considerations, making strict comparability less relevant.

A further line of argument suggests that the responsibility for reproducibility should lie primarily with software tooling rather than methodological guidelines. From this standpoint, improved libraries, standardized implementations, and automated benchmarking frameworks could mitigate many of the issues we identify without requiring explicit reporting standards. This view highlights the role of engineering and infrastructure in shaping empirical practice.

Finally, some may question whether end-to-end benchmarking is always necessary. In highly specialized settings, isolating specific components, such as the PH computation or the representation step, can provide valuable insights into algorithmic behavior that would be obscured in a full pipeline evaluation.

These alternative views underscore that the establishment of standards is not a purely technical matter but also a question of community priorities and research culture. Our recom-

mendations aim to support transparency and comparability without constraining methodological exploration, and we view them as a foundation for discussion rather than a prescriptive framework.

## 7. Conclusion

This work set out to clarify the empirical foundations of TML by examining, in a unified manner, how design choices at every stage of the pipeline affect both computational cost and the structure of the resulting topological representations. Across modalities and filtrations, our analyses revealed a simple but often overlooked fact: TML pipelines do not behave uniformly, and seemingly minor choices can lead to large variations in diagram size, representation cost and ultimately model performance. By isolating these effects through controlled experiments, scaling studies, and systematic comparisons of representations methods, we provided a concrete basis for understanding why empirical results in TML are sometimes inconsistent or difficult to reproduce. Building on these observations, we argued for the establishment of experimental standards that make TML research more transparent and comparable. These include explicit reporting of filtration and post-processing steps, systematic measurement of computation times, and the use of unified evaluation metrics supported by appropriate statistical tests. Finally, we recommend that TML pipelines be evaluated end-to-end, rather than through isolated components. End-to-end benchmarking prevents misleading conclusions that arise when only the PH computation or only the embedding step is evaluated in isolation. It also facilitates fair comparisons with non-topological methods, which are typically evaluated as complete pipelines. In conclusion, these standards aim to foster a culture of transparency, comparability, and reproducibility in TML research. By adopting them, the community can move toward more rigorous empirical practices and open the door to clearer scientific progress, more reliable benchmarking, and ultimately a deeper understanding of when and how topological methods provide genuine advantages in ML. Thus, helping TML mature into a methodology that can be adopted and trusted by the broader ML community.

### Software and Data

All experiments conducted for this article are publicly available at https://gitlab.xlim.fr/icasti02/position_icml26.

### Acknowledgements

This work and the IMASAMART study (NCT05709210) were supported by a doctoral grant co-funded by the Region Nouvelle-Aquitaine, the Poitiers University Hospital (CHU

---

[2]https://www.icmla-conference.org/icmla19/

Poitiers), and the Aliénor Foundation of Poitiers University. This work was supported by the French National Research Agency (ANR) (LabCom Damialab).

## Impact Statement

This work aims to advance the methodological foundations of Topological Machine Learning by providing a systematic analysis of empirical pipelines and their computation behavior. As a contribution focused on benchmarking, reproducibility, and methodological clarity, its primary impact lies in improving the reliability and transparency of future research in the field.

We do not identify additional societal or ethical concerns that require specific attention.

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

# A. Topological Machine Learning: Variability in the Pipeline

The following sections do not aim to provide an exhaustive survey; rather, they gather a selection of the most cited and commonly used studies in TML, acknowledging that additional relevant works may not be included.

## A.1. Datasets

To construct a representative overview of the datasets used in Topological Machine Learning, we reviewed works belonging to two major families: (i) deep-TML approaches, which integrate topological signatures within neural architectures, and (ii) supervised TML methods relying on vectorized or kernel-based representations of persistence diagrams, including recent surveys on these embeddings. The vector- or kernel-based method were chosen among the methods used in surveys (Barnes et al., 2021; Conti et al., 2022; Ali et al., 2023; Bandiziol & De Marchi, 2024). This selection ensures broad coverage of the datasets that have shaped empirical practice across point clouds, images, meshes, graphs, and time series.

The following tables summarize the benchmark datasets most frequently used in these works, organized by data modality.

*Table 5.* Benchmark datasets used in the TML literature for point clouds classification.

| DATASET | REFERENCES |
|---|---|
| ORBITS | ADAMS ET AL. (2017), CARRIÈRE ET AL. (2017; 2020), LE & YAMADA (2018), KIM ET AL. (2020), REINAUER ET AL. (2021), ROYER ET AL. (2021), CHUNG & LAWSON (2022), CONTI ET AL. (2022), BANDIZIOL & DE MARCHI (2024) |
| PCB00019 | KUSANO ET AL. (2018), POLANCO & PEREA (2019), BARNES ET AL. (2021), BANDIZIOL & DE MARCHI (2024) |
| MANIFOLDS | ADAMS ET AL. (2017), POLANCO & PEREA (2019), PEREA ET AL. (2023) |
| ANALYSIS OF $SiO_2$ | KUSANO ET AL. (2018), LE & YAMADA (2018) |
| ANALYSIS OF GRANULAR SYSTEM | KUSANO ET AL. (2018), LE & YAMADA (2018) |
| KS-EQUATION | ADAMS ET AL. (2017) |

*Table 6.* Benchmark datasets used in the TML literature for images classification.

| DATASET | REFERENCES |
|---|---|
| MNIST | ADCOCK ET AL. (2016), GARIN & TAUZIN (2019), KALIŠNIK (2019), KIM ET AL. (2020), BARNES ET AL. (2021), CONTI ET AL. (2022), BANDIZIOL & DE MARCHI (2024) |
| OUTEX_TC_00000 | REININGHAUS ET AL. (2015), CARRIÈRE ET AL. (2017), CHUNG & LAWSON (2022) |
| FMNIST | CONTI ET AL. (2022), ALI ET AL. (2023), BANDIZIOL & DE MARCHI (2024) |
| MPEG-7 | HOFER ET AL. (2017; 2019), LE & YAMADA (2018) |
| HEPATIC LESIONS | ADCOCK ET AL. (2016) |
| ANIMAL | HOFER ET AL. (2017; 2019) |
| USC-SIPI | ATIENZA ET AL. (2020) |
| HAM10000 | BARNES ET AL. (2021) |
| UIUCTEX | CHUNG & LAWSON (2022) |
| KTH-TIPS | CHUNG & LAWSON (2022) |
| OUTEX_TC_00013 | ALI ET AL. (2023) |

In Table 7, the datasets are grouped into three categories: bio-oriented molecular and protein graphs appear in the upper section, while social-networks graphs are listed in the mid section, and meshes appear in the lower section.

For time-series data, the TML literature primarily relies on benchmark datasets such as ROSSLER PERIODICITY used by Perea et al. (2023) and ECG200, SONY, DISTAL, STRAWBERRY, POWER, MOTE used by Bandiziol & De Marchi (2024).

For personal data, EEG recordings have been used in TML studies, notably in Hofer et al. (2019), to evaluate topological representations on physiological signals.

*Table 7.* Benchmark datasets used in the TML literature for graph and mesh classification.

| DATASET | REFERENCES |
| --- | --- |
| MUTAG | CARRIÈRE ET AL. (2020), REINAUER ET AL. (2021), ROYER ET AL. (2021), BANDIZIOL & DE MARCHI (2024), WANG ET AL. (2024) |
| DHFR | CARRIÈRE ET AL. (2020), ROYER ET AL. (2021), BANDIZIOL & DE MARCHI (2024), WANG ET AL. (2024) |
| PROTEINS | CARRIÈRE ET AL. (2020), ROYER ET AL. (2021), BANDIZIOL & DE MARCHI (2024), WANG ET AL. (2024) |
| COX2 | CARRIÈRE ET AL. (2020), ROYER ET AL. (2021), WANG ET AL. (2024) |
| NCI | CARRIÈRE ET AL. (2020), ROYER ET AL. (2021), WANG ET AL. (2024) |
| PTC | BANDIZIOL & DE MARCHI (2024) |
| BZR | BANDIZIOL & DE MARCHI (2024) |
| ENZYMES | BANDIZIOL & DE MARCHI (2024) |
| REDDIT | HOFER ET AL. (2017; 2019), CARRIÈRE ET AL. (2020), ROYER ET AL. (2021), WANG ET AL. (2024) |
| COLLAB | CARRIÈRE ET AL. (2020), ROYER ET AL. (2021), CONTI ET AL. (2022), WANG ET AL. (2024) |
| IMDB | CARRIÈRE ET AL. (2020), ROYER ET AL. (2021), WANG ET AL. (2024) |
| FRNKNSTN | ROYER ET AL. (2021) |
| SHREC14 | REININGHAUS ET AL. (2015), HOFER ET AL. (2019), POLANCO & PEREA (2019), BARNES ET AL. (2021), ALI ET AL. (2023), PEREA ET AL. (2023), BANDIZIOL & DE MARCHI (2024), |
| SHREC07 | DI FABIO & FERRI (2015) |
| 3D MESH SEGMENTATION | CARRIÈRE ET AL. (2017) |

## A.2. Filterings

For MNIST, we show different filtering functions applied to a single digit class (5), for different parameters. This allows to isolate the effect of the filtering choice on performance.

For height-, radial-, density-based, and Median Canny Distance filterings, the images were first binarized using a threshold of 0.4.

### A.2.1. INVERSED (I)

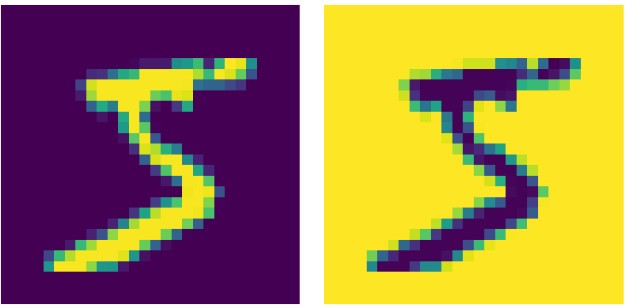

*Figure 3.* Illustration of processing (I): the left panel shows the original image used for standard filtering, while the right panel displays its inverted version used for the complementary filtering.

### A.2.2. HEIGHT (H), RADIAL (R) AND DENSITY (D)

The following filtering are described in Conti et al. (2022).

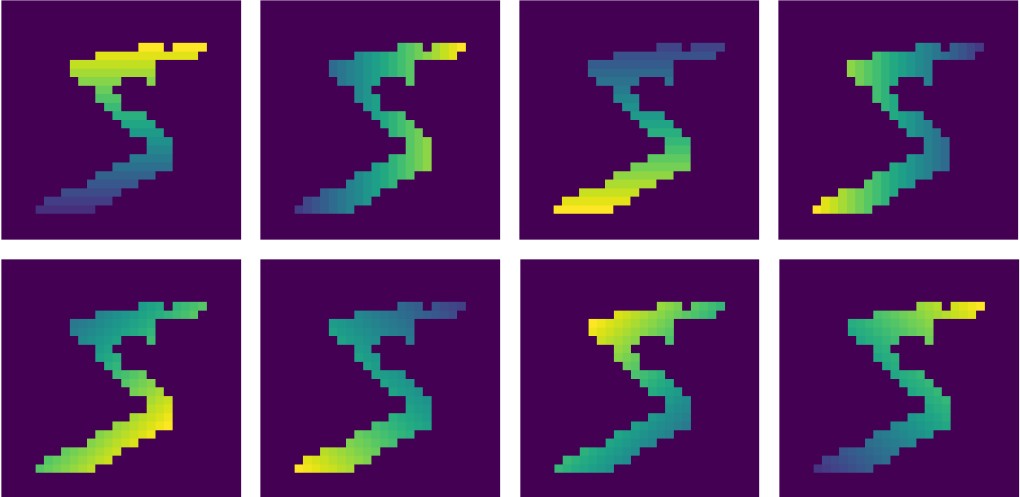

*Figure 4.* Illustration of Height filtering (H): the first row corresponds to directions oriented from bottom to top, and then, rotated clockwise. The second row shows the diagonal directions, starting from the top-left to bottom-right corners and again rotating clockwise.

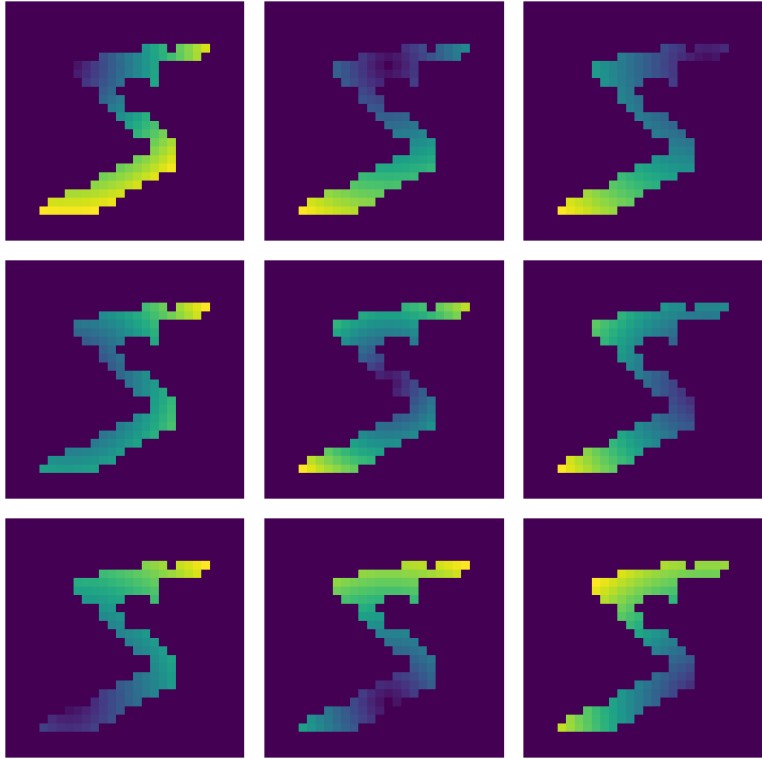

*Figure 5.* Illustration of Radial filtering (R): the first row places the center at the top-left of the grid and moves it horizontally to the right. The second row starts from the middle-left position and again shifts the center rightward. The final row follows the same pattern, with centers progressing from the bottom-left toward the right.

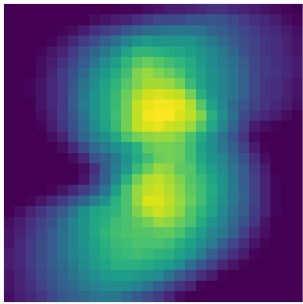

*Figure 6.* Illustration of Density filtering (D).

### A.2.3. MEDIAN CANNY DISTANCE (MCD)

The following filtering description is found in Ali et al. (2023).

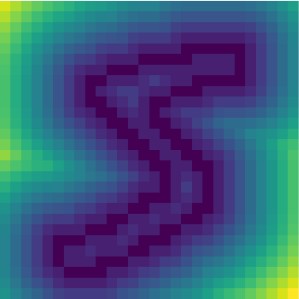

*Figure 7.* Illustration of Median Canny Distance (MCD).

### A.2.4. CLBP (GUO ET AL., 2010)

The following filtering description is found in Reininghaus et al. (2015).

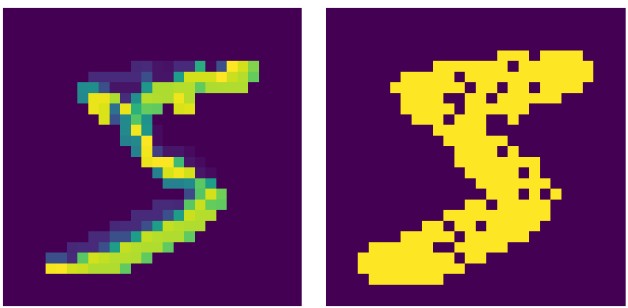

*Figure 8.* Illustration of CLBP-S (left panel) and CLBP-S (right panel).

### A.3. End-to-End Processing Methods

Table 8 summarizes the end-to-end processing methods used for images in the TML literature, combining in a unified view the filtering functions, and the post-processing steps applied to the resulting persistence diagrams. While Table 9 summarizes the end-to-end processing methods used for graph or meshes in the TML literature, combining filtering functions, and filtration.

The last column reports works that have described or adopted the method. To ensure that the comparison focuses on widely used approaches, we restrict our selection to methods applied to datasets that appear in at least three different studies.

*Table 8.* Processing pipelines for image data in the TML literature.

| METHOD | FILTERING + POST-PROC. | (EXAMPLES) |
|---|---|---|
| I | CONCATENATION OF NORMAL + INVERSED | (CHUNG & LAWSON, 2022; ALI ET AL., 2023) |
| H | CONCATENATION OF 4 HEIGHT DIRECTIONS | (ADCOCK ET AL., 2016; KALIŠNIK VEROVSEK, 2019) |
| HRD | CONCATENATION OF 8 HEIGHT DIRECTIONS + 9 RADIAL CENTERS + 1 DENSITY MAP | (CONTI ET AL., 2022) |
| MCD | SEE FIGURE 3 | (ALI ET AL., 2023; BANDIZIOL & DE MARCHI, 2024) |
| CLBP | CONCATENATION OF CLBP-S + CLBP-M | (REININGHAUS ET AL., 2015) |

*Table 9.* Processing pipelines for graph and mesh data in the TML literature.

| METHOD | FILTERING + FILTRATION | USED BY |
|---|---|---|
| HKS SUB | HKS + SUB-LEVEL SETS | (REININGHAUS ET AL., 2015; HOFER ET AL., 2019; POLANCO & PEREA, 2019; BARNES ET AL., 2021; ROYER ET AL., 2021; ALI ET AL., 2023; PEREA ET AL., 2023) |
| HKS EXT | HKS + EXTENDED PERSISTENCE | (CARRIÈRE ET AL., 2020; REINAUER ET AL., 2021) |
| GEODESIC | GEODESIC DISTANCE POINT | (CARRIÈRE ET AL., 2015; 2017) |
| 2D MATCHING | 2-DIMENSIONAL MATCHING DISTANCE | (BIASOTTI ET AL., 2011; DI FABIO & FERRI, 2015) |
| VERTEX DEGREE | DEGREE-BASED FILTRATION | (HOFER ET AL., 2017) |
| EDGE WEIGHT | WEIGHT-BASED CLIQUE FILTRATION | (CONTI ET AL., 2022) |
| WEIGHTED RIPS | SHORTEST PATH DISTANCE + RIPS | (AKTAS ET AL., 2019; BANDIZIOL & DE MARCHI, 2024) |

### A.4. Evaluation Reporting

We restrict our analysis to datasets that appear at least three times in the TML literature, and we aggregate the corresponding evaluation parameters to provide a coherent comparative overview.

A.4.1. EVALUATION METRICS, NUMBER OF RUNS AND STATISTICAL REPORTING

Across the TML literature, point cloud performance is primarily evaluated on variants of the Orbits datasets, which differ in size but share common structure of 1000 points per point cloud and 5 classes. ORBITS250 (50 individuals) is used with 10 trials in Adams et al. (2017), under a $70/30$ split in Bandiziol & De Marchi (2024) and under a $80/20$ split in Conti et al. (2022). 100 trials were used under a $50/50$ split in Chung et al. (2022). ORBITS500 (100 individuals) appears with 10 trials under a $70/30$ split in Carrière et al. (2017). ORBITS5K (1000 individuals) appears under a $70/30$ split for 5, 20 and 100 trials, in Reinauer et al. (2021), Kim et al. (2020) and Le & Yamada (2018), Carrière et al. (2020), Royer et al. (2021), respectively. Carrière et al. and Reinauer et al. did the same evaluation on ORBITS100K. For Chung & Lawson (2022) and Conti et al. (2022), the parameters values are r={2.0, 3.5, 4.0, 4.1, 4.3} , whereas all other works use r={2.5, 3.5, 4.0, 4.1, 4.3}.

The PCB00019 dataset (1357 point clouds) has 55 different classification tasks with 2 classes and fixed splits, the performance is evaluated on the average performance of the 55 tasks in Polanco et al. (2019), Barnes et al. (2021) and Bandiziol & De Marchi (2024)

The Manifolds dataset (200 points, 6 classes) is used with 10 trials and 67/33 split in Polanco et al. (2019) and Perea & Polanco (2023). Number of individuals vary, between 10 and 200 per class.

For MNIST, evaluation settings vary substantially across the literature. Adcock et al. (2016) consider between 1000 and 10000 digits, and Kališnik (2019) adopts the same ranges while specifying 100 runs, though no explicit train/test split is given in either case. Garin et al. (2019) use a 40K/10K split together with cross-validation, without reporting the number of trials. Conti et al. (2022) evaluate on a 5000/1250 split over 10 runs, whereas Bandiziol & De Marchi (2024) use 10000 digits with a 70/30 split and 10 runs. FMNIST follows a similar pattern, author that use both MNIST and FMNIST keep the same settings (Conti et al., 2022; Bandiziol & De Marchi, 2024), Ali et al. (2023) rely on the full dataset with a 70K/10K split repeated 100 times. For Outex_TC_00000, all authors use the 10 predefined 50/50 train/test splits.

For graph datasets, the evaluation configuration strongly depends on the dataset considered, as several benchmarks come with predefined experimental protocols (number of graphs, and number of graph per class). When no fixed protocol is imposed, a 70/30 train/test split is the most common choice in the literature. The number of trials varies across studies, ranging from 10 runs (Conti et al., 2022; Reinauer et al., 2021) up to 100 runs (Carrière et al., 2017).

### A.4.2. COMPUTATIONAL COSTS, ROBUSTNESS AND PERTURBATION ANALYSIS

Several works have examined computational aspects of TDA pipelines from different angles. Barnes et al. (2021) analyzed the computational costs of various vectorization methods and a kernel method as a function of dataset size, highlighting how scalability differs across representations. Chung & Lawson (2022) investigated how the runtime of vectorization methods grows with the number of points in the persistence diagram. In addition, they investigated on noise sensitivity. Le & Yamada (2018) have also reported computation times for kernel-based approaches, offering complementary insights into the efficiency of TDA feature maps.

Carrière et al. (2020) reported timing experiments to compare parameter choices and to study the relationship between computation time and predictive accuracy.

Kim et al. (Kim et al., 2020) provided an in-depth analysis of robustness to noise and corruption.

## B. Topological Machine Learning: Empirical Analysis of the Pipeline

### B.1. Dynamical System Dataset

The discrete dynamical system dataset known as the *linked twist map* (Hertzsch et al., 2007) modeling fluid flow is a continuous dynamical systems that captures the behavior of a flow by following a particle's location at discrete time intervals. The particle's trajectory, also called *orbit* is given by the discrete dynamical system :

$$
\begin{aligned}
x_{n+1} &= x_n + r y_n (1 - y_n) \mod 1, \\
y_{n+1} &= y_n + r x_{n+1} (1 - x_{n+1}) \mod 1,
\end{aligned}
\tag{1}
$$

where $r > 0$ is a parameter controlling the particle's trajectory.

For a given choice of parameter $r > 0$ and an initial condition $(x_0, y_0) \in [0, 1]^2$, the linked twist map generates an orbit $X = \{x_i\}_{i=1}^N \subset \mathbb{R}^2$.

Depending on the choice of $r$, the orbits can exhibit either dense trajectories or voids, making this dataset particularly suitable for testing the sensitivity of topological data analysis methods. In practice, truncated orbits with finite $N \in \mathbb{N}$ are used to generate point sets whose topological signatures reflect the underlying dynamics.

### B.2. Image Filtering Analysis

We refer to the filtering configurations introduced in Section A.2, using the abbreviations t=top, b=bottom, r=right, l=left, and m=middle, where the symbol "-" denotes the direction of the filtering ($\rightarrow$).

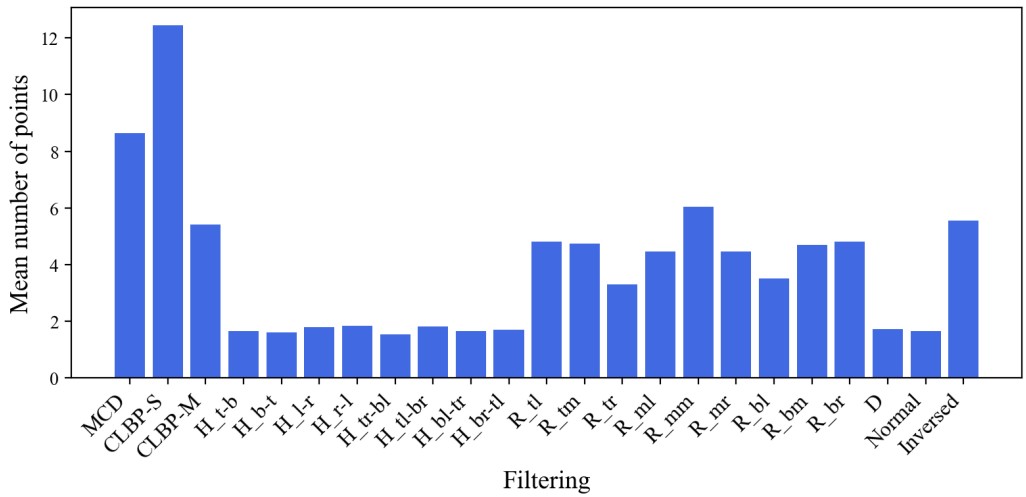

*Figure 9.* Average number of $H_0$ points obtained for each filtering strategy on the MNIST dataset.

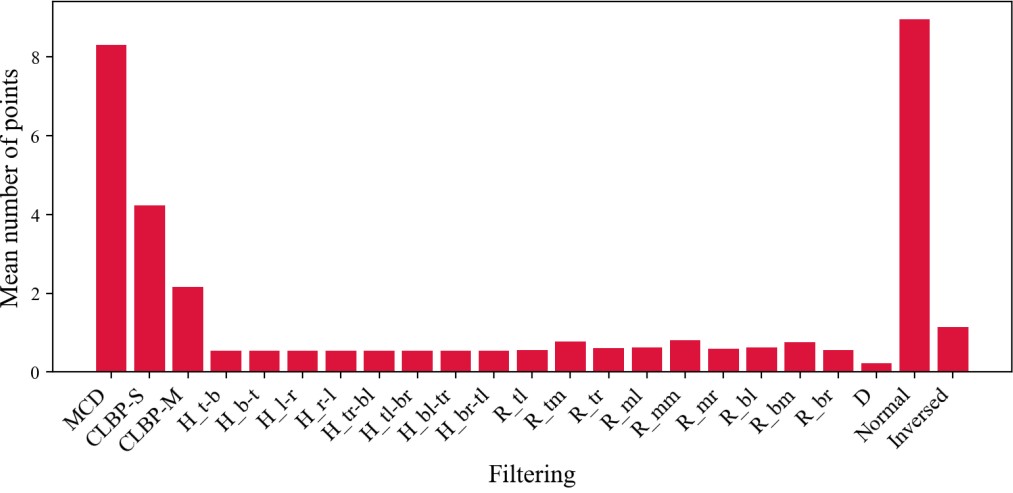

*Figure 10.* Average number of $H_1$ points obtained for each filtering strategy on the MNIST dataset.

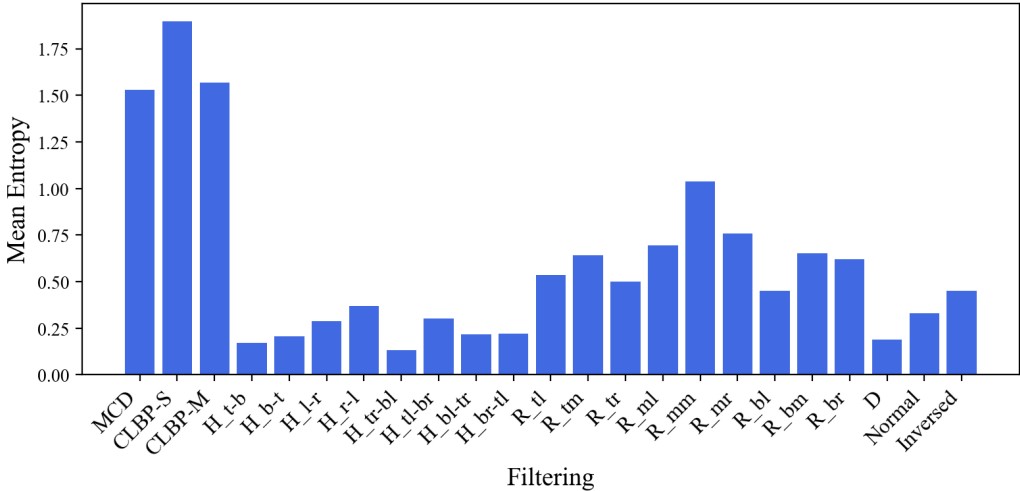

*Figure 11.* Average Entropy of $H_0$ points obtained for each filtering strategy on the MNIST dataset.

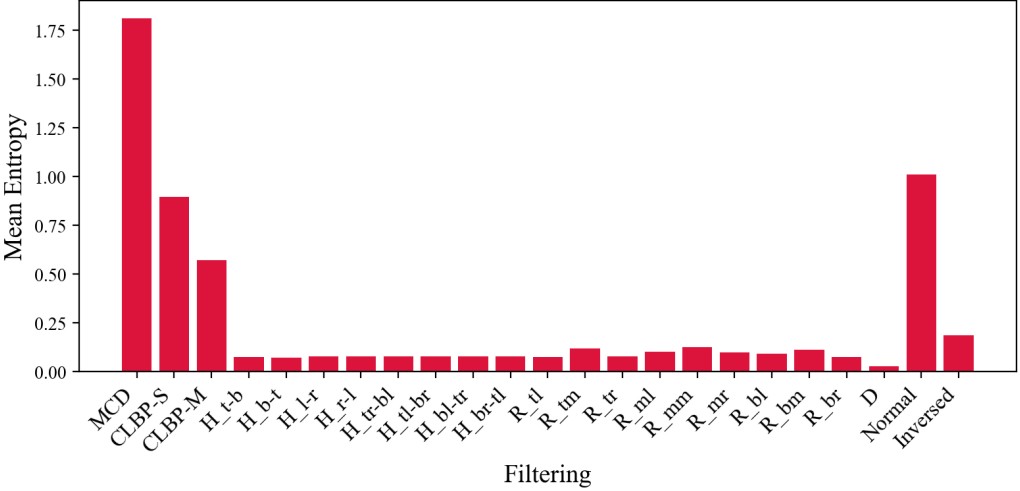

*Figure 12.* Average Entropy of $H_1$ points obtained for each filtering strategy on the MNIST dataset.

