# OpenReview forum: "Position: Topological Machine Learning Cannot Progress without Experimental Standards"
_ICML.cc/2026/Position_Paper_Track — ICML 2026 Position Paper Track regular_

### Official Review · Reviewer_QEpV · 2026-03-01

**Significance:** 3
**Argument Clarity:** 3
**Rating:** 4
**Confidence:** 2

**Questions:**

See weaknesses.

**Alternative Views Section:**

Yes

**Compliance With Llm Reviewing Policy A Conservative:**

Affirmed.

**Discussion Potential:**

3

**Final Justification:**

My concerns have been successfully resolved, and I recommend accepting this paper.

**Paper Summary:**

This paper argues that current research in Topological Machine Learning (TML) suffers from a lack of unified experimental standards, making methods difficult to compare and replicate. By analyzing the design variability across the TML pipeline and experiments on point clouds, images, and mesh data, the paper demonstrates that these seemingly minor choices significantly impact the complexity and computational cost of persistence diagrams. To address this, the authors advocate for the establishment of clear experimental standards, including end-to-end pipeline reporting, comprehensive computation time statistics, and unified evaluation metrics with statistical tests, to enhance the transparency and comparability of TML research.

**Position:**

Yes

**Position In Title:**

Yes

**Related Work:**

3

**Strengths And Weaknesses:**

Strengths:

1. This paper identifies an important point in the current TML field: the high heterogeneity in experimental designs makes results impossible to compare and reproduce. The problem statement is clear and timely.
2. The analysis is systematic and in-depth. The authors quantify the impact of different design choices through experiments on multimodal data, strengthening the persuasiveness of the argument and providing a concrete basis for the subsequent formulation of standards.
3. The writing is easy to follow.

Weaknesses:

1. The formation of community consensus may require organizational efforts (e.g., TML workshops), but the paper does not discuss a specific path for implementation.
2. Reporting computation times alone might not be sufficient to depict the computational cost, as many factors (e.g., hardware) may significantly impact timing.
3. As acknowledged in the Alternative Views section, premature or overly rigid standards may stifle methodological diversity, particularly in a field like TML that is still rapidly evolving.

**Support:**

3

---

> ### Author Rebuttal · Authors · 2026-03-30
>
> Dear reviewer,
>
> We thank you for the thoughtful and encouraging assessment of our work. Please feel free to follow up if you have further questions.
>
> **Question 1: On the need for community consensus and implementation pathways**
>
> We agree that establishing standards would require collective effort. While this paper focused on identifying methodological issues and proposing concrete reporting guidelines, we will clarify in the revised version that implementation naturally depends on community-driven initiatives. The revised version would have a sub-section:
>
> **5.3 Implementation Pathways**
>
> Several community initiatives have already demonstrated the value of coordinated efforts around TDA and its interaction with ML. Notable examples include the Topological Data Analysis and Beyond workshop at the 2020 NeurIPS conference (https://tda-in-ml.github.io/), as well as the Special Session on Topological Data Analysis in Machine Learning held at the 18th IEEE International Conference on Machine Learning and Applications (ICMLA 2019) (https://www.icmla-conference.org/icmla19/). Similar Workshops or Special Sessions could be organized in the upcoming years to reflect on the evolution of TDA within modern ML pipelines (TML): integrating (i) experimental standards, (ii) insights from recent surveys, (iii) dialogue around deep-TML approaches. Such recurring community-driven efforts would naturally support the development and adoption of shared experimental standards, and further facilitate the integration of TML in the broader ML community.
>
> **Question 2: On the limitations of reporting only computation times**
>
> We agree that raw computation times alone do not fully capture computational cost, since hardware characteristics, implementation details, and parallelization can significantly influence timing. This
> reinforces the need for transparent reporting of hardware and software specifications in order to put computation times into perspective (as we did in **Section 4**). The main focus needs to be on the *comparison of computation times* rather than on an *absolute computation time*.
>
>
> **Question 3: On the risk of premature or overly rigid standards**
>
> As noted in the Alternative Views section, we fully acknowledge that methodological diversity is essential for a field as young and rapidely evolving as TML. Our intention is not to restrain innovation, but ensure interpretable and comparable experimental results.
> At the same time, the absence of minimal standards currently slows down the growth of the field: many TML papers face long delays before publication, often remaining on preprint servers for months or years and some never gain visibility. In our opinion, the lack of shared experimental standards contributes to the difficulty of evaluating, comparing, and ultimately adopting new methods.

---

> > ### Author Rebuttal · Reviewer_QEpV · 2026-04-02
> >
> > Thanks for the rebuttal. I will maintain my positive score.

---

### Official Review · Reviewer_qnga · 2026-03-10

**Significance:** 3
**Argument Clarity:** 3
**Rating:** 5
**Confidence:** 4

**Questions:**

Can you highlight examples in which the statistical significance of performance is assessed appropriately?

It is stated that the longer a feature persists across scales, the more it is considered topologically significant. Is that always the case?

**Alternative Views Section:**

Yes

**Compliance With Llm Reviewing Policy A Conservative:**

Affirmed.

**Discussion Potential:**

3

**Paper Summary:**

The paper points out that current applications of topological data analysis in machine learning (TML) often lack detailed descriptions of procedures such as choice of filtration, computing time, and statistical evaluation. It advocates for a standardised experimental framework for TML which includes descriptions of
* the data set and sysnthetic data constructions
* the filtering functions and their parameters
* the structure type
* any simplifications or aggregations
* the embedding method.
It also recommends that comparisons should only be made between identical pipelines.

**Position:**

Yes

**Position In Title:**

Yes

**Related Work:**

2

**Strengths And Weaknesses:**

The paper gives a long list of TML applications in the appendix. It also details effects of different choices on the outcome.

The alternative view section is sensible and a good basis for initiating discussion.

It would have been good to see some mention of theoretical underpinnings. For example

Bobrowski, Omer, and Primoz Skraba. "A universal null-distribution for topological data analysis." Scientific reports 13.1 (2023): 12274

give a universal limit theorem for persistence diagrams. Results of this type should be reflected in the reporting requirements.

Moreover point cloud data are often not i.i.d. and hence standard statistical frameworks (such as t-tests) may not apply, It would have been good to see the need to develop appropriate statistical methods discussed in more detail.

**Support:**

3

---

> ### Author Rebuttal · Authors · 2026-03-30
>
> Dear reviewer,
>
> Thank you very much for taking the time to review our submission! Please feel free to follow up if you have further questions.
>
> We agree that the reference suggested by the reviewer Bobrowski, Omer, and Primoz Skraba. "A universal null-distribution for topological data analysis." Scientific reports 13.1 (2023) is relevant. Their universal limit theorem provides a principled description of the distribution of persistence diagrams, where the filtration is built directly from the *point cloud* (e.g., Vietoris-Rips, Čech). This is particularly important in settings where no filtering function is applied. In our context, however, many TML pipelines rely on design choices (filtering, structure) that have an effect on the distribution of points in the persistence diagram. Our conclusions, supported by our experiments, are: (i) design choices affect the distribution of points in the persistence diagram, (ii) this distribution is then either used directly by deep TML methods or is transformed by supervised TML methods, (iii) meaning that two methods X and Y built on different design choices operate on *different distributions*. This reinforces our central claim: to ensure meaningful comparability between methods X and Y, the entire pipeline should be identical, so that differences in performance can be attributed to the method itself rather than design choices. In the revised version, we will add a brief remark on how our hypothesis might direclty be linked to the distribution of points(+citation) in the persistence diagrams, rather than just the "structure" of persistence diagrams. Further work on statistical tools for persistence diagrams would, in this case, be highly beneficial.
>
> **Q1 : Examples of appropriate statistical assessment ?**
>
> To the best of our knowledge, *no existing TML study evaluates the statistical significance of performance differences*. Classical studies, in ML, are referrenced (Dietterich, 1998 and Demsar,  2006), they already provide frameworks for appropriate statistical assessment. These methodologies transfer naturally to TML. Following the reviewer's insightful comments and questions, the **Section 5.2** (line 390 to 396) will be reformulated as follows:
>
> Given two methods TML methods, classical tests include the McNemar test to test error patterns (citation Dietterich, 1998), and paired t-test or none-parametric Wilcoxon signed-rank test on performance distributions (citation Demsar, 2006).
>
> **Q2 : Are long-persistence features always more significant?**
>
> In TDA, points near the diagonal ("short-persistence" features) are classicaly considered as topological noise. On the contrary, "long-persistence" features capture holes. As an example, long-persistence features charaterize classical topolopgical spaces like: ball, sphere, torus with $n$ holes...

---

> > ### Author Rebuttal · Reviewer_qnga · 2026-04-02
> >
> > Thank you for your reply. As far as I recall there is some discussion in TDA about the importance of intermediate bar code length; it may be prudent to make clear that it is currently thought that long-persistence features are more significant.

---

### Official Review · Reviewer_uDJD · 2026-03-13

**Significance:** 3
**Argument Clarity:** 3
**Rating:** 5
**Confidence:** 3

**Questions:**

1. How do design choices affect downstream accuracy?

**Alternative Views Section:**

Yes

**Compliance With Llm Reviewing Policy A Conservative:**

Affirmed.

**Discussion Potential:**

3

**Final Justification:**

My questions have been addressed by the rebuttal and have reinforced my prior positive rating. To avoid the use of a borderline rating, I will increase my score to Accept.

**Paper Summary:**

This paper takes the position that progress in topological machine learning (TML) depends on establishing clear experimental standards that allow the easy comparison of different methods. This paper surveys and gathers data on each step of the pipeline where there are choices to be made, and shows that there is a wide variety in datasets used, metrics used, filtering functions used, and complex used and how these details are reported. The paper also runs experiments that show how different choices affect the computational efficiency and the generated persistence diagrams. Based on this, the paper proposes a set of experimental standards for papers to adhere to.

**Position:**

Yes

**Position In Title:**

Yes

**Related Work:**

3

**Strengths And Weaknesses:**

Strengths:

I believe that this paper raises an important point regarding the state of TML research and that this paper will likely inspire good discussion on the choice of benchmarks and metrics for TML. This paper also has a clear call to action, and in general clarity is also a strength of this paper. I believe that the alternative views section also presents multiple credible viewpoints.

Weaknesses:

While I am convinced by the evidence that changes in pipeline can result in large changes in computation time and persistence diagram structure, I think that it may also be helpful to show the effect of some of these choices on downstream accuracy.

**Support:**

3

---

> ### Author Rebuttal · Authors · 2026-03-30
>
> Dear reviewer,
>
> Thank you very much for taking the time to review our submission! We are pleased to see your encouraging points about our position being clear, and up to good discussion. Please feel free to follow up if you have further questions.
>
> **Q1: How does design choices affect downstream accuracy?**
>
> In addition to the computational and structural variability in persistence diagrams (highlighted in the article), we also investigated how design choices affect *downstream accuracy*. We conducted additional experiments on the three datasets (orbits, mnist and shrec) using two vectorization methods that do not involve hyperparameters. This choice was intentional: by relying on parameter-free vectorizations, we avoid confounding effects due to parameter optimization and isolate the influence of the design choices.
>
> Across datasets, we compared multiple design choices (filtering, structure, post-processing). Even under this controlled setting, we observed substantial variability in downstream accuracy: *both vectorization methods produced markedly different performance results*, and *best accuracy was not consistently associated with any specific design choices*. Results are reported in the Table below; it reports mean accuracy of fixed Random Forests (100 trees) over 100 runs on specific 70/30 splits.
>
> |       	|       	    |  Persistence  | Statistics	|       	 	|               |  	Carlsson    |   Coordinates |               |
> |-----------|---------------|---------------|---------------|---------------|---------------|---------------|---------------|---------------|
> |       	|    $H\_0$ 	|    $H\_1$    	|  $H\_0+H\_1$ 	|   $H\_0H\_1$ 	|    $H\_0$ 	|    $H\_1$  	|  $H\_0+H\_1$  |   $H\_0H\_1$ 	|
> |**orbits** |       	    |               |               |       	 	|               |  	            |               |               |
> | Alpha 	|     81.85 	|    88.15    	|  **88.53** 	|    87.93  	|    74.64 	    |    78.96  	|    78.33    	|    75.87  	|
> | Rips  	|     83.24 	|    87.31    	|    86.52   	|    84.60  	|    74.24 	    |  **88.43** 	|    86.76    	|    83.67  	|
> |**mnist**  |       	    |               |               |       	 	|               |  	            |               |               |
> | H 		|    61.17  	|    76.21    	|    82.95 	    |    83.91  	|    42.79  	|    79.17  	|  **85.99**  	|    82.83  	|
> | HRD  		|    67.30  	|    84.19    	|  **86.87**  	|    77.25  	|    74.24  	|    80.86  	|    84.06    	|    83.93  	|
> |**shrec14**|       	    |               |               |       	 	|               |  	            |               |               |
> | 1 		|    70.36  	|    79.03    	|  **80.54** 	|    76.87  	|    50.29 	    |    43.94  	|    60.14  	|    44.18  	|
> | 5  		|    37.27 	    |    88.02    	|    87.67  	|    86.66  	|    15.52 	    |  **88.53**	|    87.62    	|    88.04  	|
> | 10  		|    51.90 	    |  **88.19**    |    86.49  	|    87.18  	|    49.99 	    |    63.39  	|    61.72    	|    63.56  	|
>
> Table : Mean accuracy for different design choices on the orbits, mnist and Shrec14 Dataset rescpectively.
>
> We also performed statistical tests to *identical design choices*, resulting in *93.75% of p-values < 0.05*. These tests confirm that, even when the processing steps are the same, accuracy differences between vectorization methods remain statistically significant in many cases. However, for the same dataset and fixed classifier/splits, best accuracy varies with the chosen pipeline, and no design choices emerge as universally optimal.
>
> This corroborates our position that control over the experimental setup and design choices is essential to draw reliable conclusions about performance of a given TML method. Overall, these results show that design choices in TML *do affect* downstream accuracy. This further motivates our Position on the need for more standardized, transparent, and reproducible experimental practices in TML.
>
> In response to the reviewer's question, we acknowledge that reporting these empirical results could strengthen our Position. Therefore, we plan to include a section (between the Call to Action and Alternative Views sections) that shows the results above.

---

> > ### Author Rebuttal · Reviewer_uDJD · 2026-04-02
> >
> > Thanks for the response. I encourage the addition of these experiments to the paper if possible.

---

### Decision · Program_Chairs · 2026-04-30

**Decision:**

Accept (regular)

**Comment:**

This paper makes a strong position-paper contribution. It identifies a genuine obstacle to progress in topological machine learning: experimental heterogeneity across the full TML pipeline makes results difficult to compare, reproduce, and interpret. The paper empirically demonstrates that design choices substantially affect persistence-diagram size, entropy, and computation time across point clouds, images, and meshes, and the rebuttal further strengthens the case by showing that these choices also affect downstream accuracy. The proposed standards are concrete, moderate, and actionable rather than overly prescriptive.

The remaining weaknesses are limited. It would be useful to connect the position more explicitly to theoretical statistical tools for persistence diagrams, to discuss a bit more carefully when classical significance tests may fail, and to say more about how community adoption of standards might be organized. However, these are extensions, not fundamental gaps. The core argument is clear, well supported, and well aligned with the goals of the track.

The recommendation is accept. The paper identifies an important methodological need, supports it with concrete evidence, and offers a practical framework that could help make future TML research more trustworthy and easier for the broader ML community to evaluate.